# Limitations of Normalization in Attention Mechanism

**Timur Mudarisov**
University of Luxembourg
Luxembourg
timur.mudarisov@uni.lu

**Mikhail Burtsev**
London Institute for
Mathematical Sciences
mb@lims.ac.uk

**Tatiana Petrova**
University of Luxembourg
Luxembourg
tatiana.petrova@uni.lu

**Radu State**
University of Luxembourg
Luxembourg
radu.state@uni.lu

## Abstract

This paper investigates the limitations of the normalization in attention mechanisms. We begin with a theoretical framework that enables the identification of the model's selective ability and the geometric separation involved in token selection. Our analysis includes explicit bounds on distances and separation criteria for token vectors under softmax scaling. Through experiments with pre-trained GPT-2 model, we empirically validate our theoretical results and analyze key behaviors of the attention mechanism. Notably, we demonstrate that as the number of selected tokens increases, the model's ability to distinguish informative tokens declines, often converging toward a uniform selection pattern. We also show that gradient sensitivity under softmax normalization presents challenges during training, especially at low temperature settings. These findings advance current understanding of softmax-based attention mechanism and motivate the need for more robust normalization and selection strategies in future attention architectures.

## 1 Introduction

The attention mechanism [1, 7, 17, 8] has become a fundamental component of modern deep learning. Since its popularisation in the Transformer [18], attention has powered state-of-the-art systems in machine translation, text generation [3], and multimodal reasoning [11]. Yet the same softmax rule that enables differentiable "focus" also introduces a chronic failure mode: as the context length $L$ grows, attention weights collapse toward $1/L$, a phenomenon we call *vanishing attention*. The resulting gradients are too small for effective learning, especially in long-context settings [4, 21]. Architectural work-arounds – sparse windows, locality-sensitive hashing, or compressed memories—reduce compute but do not eliminate the collapse, and may even worsen it [5, 16, 2, 10]. A principled understanding of *why* softmax fails in this regime is still missing.

**Our view: attention as a capacity-limited retriever (selector).** We revisit normalisation in attention from first principles and show that softmax, and indeed *any* length-independent normaliser, possesses an intrinsic capacity limit. Our contributions are:

1. **Distance bound.** We derive a non-asymptotic upper bound on the representation distance between selected and non-selected tokens (Theorem 1). The bound proves that once the top-$N$ set grows proportionally to $L$, the distance *must* collapse, formalising the "softmax bottleneck" [20].

2. **Geometric bound.** Under mild spherical assumptions we show that no more than $\approx 80\%$ of the top-$N$ tokens can be simultaneously distinguished in Euclidean space (Theorem 2). This quantifies a hard limit on what a single head can represent.

3. **Gradient bound.** We bound the Jacobian norm of a general normaliser (Lemma 2); specialised to softmax it recovers the classic $1/(4T)$ instability and shows why aggressive temperature scaling trades separability for optimisation difficulty.

4. **Empirical validation.** Experiments on GPT-2 [15] confirm all three predictions: distance collapse, separability saturation, and $1/T$ gradient growth.

Our analysis frames attention as a *selector with finite resolution*: it works well while the active set is a small fraction of the context, then degrades predictably. This viewpoint explains the empirical success of recent alternatives—Sparsemax, Scalable-Softmax, Self-Adjusted Softmax—and suggests concrete design rules we distil in the discussion. More broadly, our theoretical tools provide diagnostics for deciding when a head has reached its intrinsic limit and when architectural or normalisation changes are warranted.

By bridging closed-form theory with large-scale experiments, we provide both a deeper understanding of normalisation in attention and practical guidelines for building robust, long-context Transformer models.

## 2 Theoretical Analysis

Consider a sequence of token embeddings $\mathbb{X} = \{x_i\}_{i=1}^L$, where each embedding $x_i \in \mathbb{R}^d$ is a $d$-dimensional vector. We begin by reviewing the classical self-attention mechanism introduced by Vaswani et al. [18]:

$$\mathbf{q}_m = f_q(x_m, m), \quad \mathbf{k}_n = f_k(x_n, n), \quad \mathbf{v}_n = f_v(x_n, n), \tag{1}$$

where $\mathbf{q}_m$, $\mathbf{k}_n$, and $\mathbf{v}_n$ denote the query, key, and value vectors, respectively. The attention weights are computed using these query and key vectors as follows:

$$a_{m,n} = \exp\left(\mathbf{q}_m^\top \mathbf{k}_n / T\right) / \sum_{j=1}^L \exp\left(\mathbf{q}_m^\top \mathbf{k}_j / T\right), \tag{2}$$

where the parameter $T$ is known as the temperature, typically set to $T = \sqrt{d}$ (as recommended in [18]). To extend the scope of our analysis beyond the standard softmax normalization, we introduce a more general normalization framework:

$$a_{m,n} = F(\mathbf{q}_m^\top \mathbf{k}_n, \theta) / \sum_{j=1}^L F(\mathbf{q}_m^\top \mathbf{k}_j, \theta), \tag{3}$$

where $F : \mathbb{R}^{1+c} \to \mathbb{R}$ is a smooth positive function parameterized by $\theta$, which can include parameters such as temperature or the number of tokens. For convenience, we denote the inner product $\mathbf{q}_m^\top \mathbf{k}_n$ as $l_{m,n}$, referring to it as the *logit* associated with the token pair $(m, n)$.

In this work, we provide a detailed theoretical examination of the general normalization framework introduced in Equation (3). We focus on several critical aspects: (1) general limitations associated with softmax-type normalization; (2) the influence of normalization on token separation; (3) geometric insights into token separation; (4) connections between normalization and training dynamics.

We begin by addressing the general limitations of softmax normalization. As highlighted by [19], one critical limitation is the phenomenon of *vanishing attention weights*. Intuitively, when the number of tokens $L$ increases, the normalization procedure (e.g., softmax) distributes attention weight across many tokens, causing the weights for individual tokens to become extremely small. This issue hampers the model's ability to clearly differentiate between relevant and irrelevant tokens.

Formally, consider a set of logits $\{l_1, \ldots, l_L\}$ and their corresponding attention weights $\{\alpha_1, \ldots, \alpha_L\}$, computed from the logits using a normalization function. We present the following general results regarding these normalized attention weights (see Appendix A for full proofs):

**Lemma 1.** *Consider the normalization scheme defined by Equation* (3) *with a smooth function* $F(l_i, \theta)$ *that does not explicitly depend on the number of tokens $L$ (i.e., $L \notin \theta$). Assume also that the logits are bounded, $l_i \in [-a, a]$. Then, the normalized attention weights satisfy:*

$$\frac{C_1}{L} \le \alpha_i \le \frac{C_2}{L}, \tag{4}$$

*where the constants $C_1$ and $C_2$ do not depend on $L$.*

This lemma implies that for any normalization of function independent of token count, the attention weights inevitably become uniformly small (on the order of $1/L$) as the context size grows. As a result, the mechanism loses the ability to effectively highlight specific important tokens when processing long sequences. The proof relies on the boundedness and continuity of the function $F(l_i, \theta)$ on the compact interval $[-a, a]$. Due to these properties, both the numerator and denominator are bounded by constants independent of $L$. Hence, when normalized by the summation over all $L$ tokens, each weight naturally scales as $1/L$. Full mathematical details are provided in Appendix A.

**Corollary 1.** *In the special case of softmax normalization (Equation* (2)*), with temperature parameter $T$, the attention weights satisfy:*

$$\frac{1}{L} \exp\left(-\frac{2\Delta}{T}\right) \le \alpha_i \le \frac{1}{L} \exp\left(\frac{2\Delta}{T}\right), \tag{5}$$

*where $\Delta = \|\mathbf{q}\|_2 \|\mathbf{k}\|_2$.*

This corollary highlights how the magnitude of the query and key vectors directly impacts the distribution of attention weights. Specifically, the bounds show that unless the vectors have sufficiently large magnitudes (relative to temperature $T$), the attention weights remain close to uniform. Thus, increasing vector magnitudes can partially mitigate—but not eliminate—the vanishing attention problem.

These results demonstrate a fundamental limitation of softmax-type normalization: as the sequence length $L$ grows, each attention weight $\alpha_i$ shrinks toward $O(1/L)$. In turn, the model struggles to assign appreciably larger weights to genuinely informative tokens, blurring the distinction between relevant and irrelevant embeddings. This loss of focus impairs the model's ability to exploit salient information within long input sequences and ultimately hampers effective learning.

## 2.1 Distance analysis

We next study how normalization influences the *total deviation*—the discrepancy between a learned representation and the underlying conditional distribution. Yang et al. [20] describe the *softmax bottleneck*: the low rank of the logit matrix limits the model's ability to represent the true conditional distribution, thereby increasing total deviation. This deviation is not a consequence of insufficient model capacity or sub-optimal optimization, but rather of restrictions imposed by the final softmax projection layer.

To quantify the role of normalization in separating informative from non-informative tokens, let $\{\alpha_1, \ldots, \alpha_L\}$ and $\mathbb{X} = \{x_1, \ldots, x_L\}$, $x_i \in \mathbb{R}^d$ denote the attention weights and their corresponding token embeddings, respectively. We focus on the top-$N$ tokens with the largest weights; their indices are collected in $I_N = \{i_1, \ldots, i_N\}$. The *context vector* derived from these tokens is

$$s = \sum_{i \in I_N} \alpha_i \, x_i. \tag{6}$$

Our quantity of interest is the cumulative distance between this context vector $s$ and all *non-selected* embeddings:

$$\tilde{d} = d(\mathbb{X} \setminus \mathbb{X}_{I_N}, s) = \sum_{i \in I \setminus I_N} \|\alpha_i \, x_i - s\|_2, \tag{7}$$

where $I = \{1, \ldots, L\}$, $\|\cdot\|_2$ is the Euclidean norm, and $\mathbb{X}_{I_N} = \{x_{i_1}, \ldots, x_{i_N}\}$ is the set of selected embeddings. Intuitively, a smaller value of $\tilde{d}$ indicates that many low-weight tokens lie close to the high-weight aggregate $s$, signalling reduced separability and a greater likelihood of attention "dilution."

Next, we distinguish two settings for the weight vector $\{\alpha_i\}$. In the first, the weights are treated as *fixed*; in the second, we analyse a *random-selection* scenario in which the index set $I_N$ is drawn uniformly at random from all subsets of size $N$: $I_N \sim U(I, N)$, $\quad I = \{1, \ldots, L\}$. In this random case, the attention weights on the non-selected tokens vary with the draw of $I_N$. We seek the expected cumulative distance

$$E = \mathbb{E}_{I_N}[\tilde{d}], \tag{8}$$

where the expectation is taken over the uniform choice of $I_N$.

We now adopt a classical metric-learning perspective: the above quantities quantify the Euclidean "separation margin" between selected and non-selected token embeddings. The next theorem formalises these observations by providing explicit upper bounds on the distance $\tilde{d}$ in both the fixed and random top-$N$ settings.

**Theorem 1.** *For the representation distance defined in* (7) *the following bounds hold.*

1. ***Fixed top-$N$ set.*** *If $I_N$ is fixed,*

$$\tilde{d} \leq (1 - \bar{\alpha}_N) d_1 + \max_{j \in I_N} \|x_j\|_2 \left[ \bar{\alpha}_N (L - N) - (1 - \bar{\alpha}_N) \right],$$

*where $d_1 = \max\limits_{i \notin I_N,\, j \in I_N} \|x_i - x_j\|_2$ and $\bar{\alpha}_N = \sum\limits_{i \in I_N} \alpha_i$.*

2. ***Uniformly random top-$N$ set.*** *When $I_N$ is sampled uniformly from all $\binom{L}{N}$ subsets of size $N$,*

$$E = \frac{L - N}{L} \sum_{i=1}^{L} \left\| (\alpha_i + \tfrac{N}{L-1}) x_i - \bar{x} \right\|_2 + \varepsilon,$$

*where $\bar{x} = \sum\limits_{i=1}^{L} \alpha_i x_i$ and $\varepsilon \leq \frac{1}{2}\left(1 - \tfrac{N}{L}\right) \sum\limits_{i=1}^{L} \frac{N}{L-1} \frac{\sum_{j \neq i} \alpha_j^2 \|x_j\|_2^2}{\left\| \alpha_i x_i - \tfrac{N}{L-1} \sum_{j \neq i} \alpha_j x_j \right\|_2}.$*

Intuitively, for a *fixed* top-$N$ set, the first term scales the largest out-of-set distance $d_1$ by the total weight $(1 - \bar{\alpha}_N)$ carried by the remaining $L - N$ tokens, while the second term accounts for how much those low-weight tokens can still perturb the context vector, proportional to the norm of the largest selected embedding. When the top-$N$ indices are chosen *uniformly at random*, each token is excluded with probability $(L - N)/L$. Replacing the random indicator variables by their expectations yields the main sum; the residual $\varepsilon$ captures the Jensen gap and is small whenever the individual weights $\alpha_i$ are not too concentrated. Full derivations are provided in Appendix A.

Theorem 1 relates the distance $\tilde{d}$ to key parameters ($N$, $L$, and the weights $\alpha_i$). To see how the bound behaves in two extreme regimes, we state the following corollary.

**Corollary 2.** (i) ***Fixed $N \ll L$.*** *When $N$ is held constant and $L$ grows,*

$$E \approx \sum_{i=1}^{L} \|\alpha_i x_i\|_2.$$

(ii) ***Fixed $L$, $N \to L$.*** *If $L$ is fixed and the top set expands to the full sequence, i.e. $N \to L$,*

$$E \longrightarrow 0. \tag{9}$$

When $N$ is small relative to the sequence length, most tokens are excluded and the expected distance is dominated by the individual contributions $\|\alpha_i x_i\|_2$ of those low-weight tokens. In contrast, as $N$ approaches $L$, the context vector $s$ eventually incorporates *all* embeddings, so the distance between $s$ and the (now empty) set of non-selected tokens vanishes.

## 2.2 Geometrical interpretation

We now recast the analysis in geometric terms, focusing on the spatial arrangement of the token embeddings $x_i$. When these vectors lie close together, the model struggles to separate informative from non-informative tokens, hindering training. To quantify how many tokens can be reliably distinguished, we work under two standard geometric assumptions:

**Assumption 1 (Uniform spherical distribution).** Each embedding lies uniformly on a $d$-dimensional sphere of radius $M$:

$$x_i \sim U\big(\mathbb{S}^{d-1}(M)\big), \qquad i = 1, \dots, L,$$

where $\mathbb{S}^{d-1}(M)$ denotes the $(d-1)$-sphere of radius $M$. In practice, we *normalize* embeddings so that they lie on this sphere.

**Assumption 2 (Minimum pairwise separation).** There exists a fixed lower bound on the distance between any two embeddings:

$$\min_{i \neq j} \|x_i - x_j\|_2 = \delta > 0,$$

ensuring that no two tokens collapse onto one another. In experiments, we set $\delta$ to the empirical minimum pairwise distance.

Under these assumptions, we can bound the number of embeddings that fall within a specified neighbourhood of the context vector, thereby quantifying the model's effective resolution.

Let $I_N = \{i_1, \dots, i_N\}$ denote the indices of the $N$ selected tokens and recall the context vector

$$s = \sum_{i \in I_N} \alpha_i x_i.$$

Fix a tolerance radius $r > 0$ and consider the closed Euclidean ball $B_r(s)$ centred at $s$. We say that a selected embedding $x_i$ is *geometrically distinguishable* if its rescaled version $\alpha_i x_i$ remains within $B_r(s)$ while every non-selected embedding satisfies $\alpha_j x_j \notin B_r(s)$ for $j \notin I_N$. The count of such distinguishable embeddings is

$$N_s = \#\big\{ i \in I_N : \|\alpha_i x_i - s\|_2 \leq r \big\}, \tag{10}$$

and the ratio $N_s/N$ measures the fraction of "non-noise" vectors the model can reliably separate (see Fig. 1 for an illustration).

Our objective is to bound the expectation $\mathbb{E}[N_s]$ and hence provide explicit upper and lower limits on the model's effective resolution as a function of $r$, $N$, and $L$ under Assumptions 1–2.

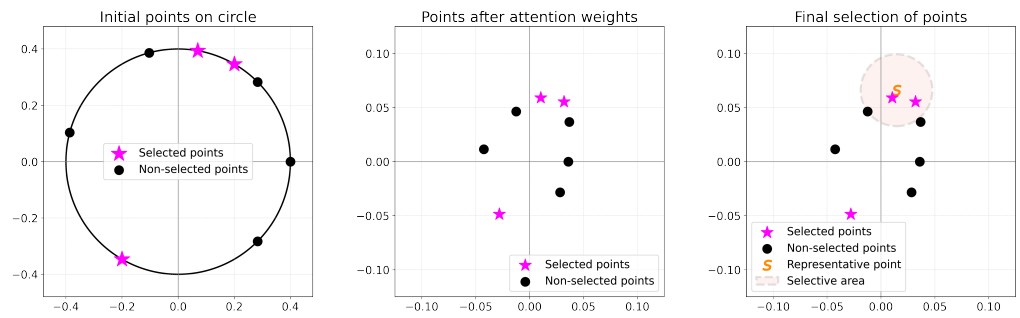

Figure 1: Illustrative example of the geometric separation. **Left:** Token embeddings lie on a circle. **Middle:** After scaling by their attention weights $\alpha_i$, both attended (magenta stars) and non-attended (black dots) points move toward the origin. **Right:** Only the selected tokens that remain inside the ball $B_r(s)$ (shaded) are deemed distinguishable.

In the previous section, we examined separation in terms of Euclidean distances. The geometric approach introduced here, by contrast, focuses on directional separability and reframes the problem as one of metric learning. Now, we present the main result (for the proof, see Appendix A):

**Theorem 2.** *Under Assumptions 1–2, the fraction of geometrically distinguishable embeddings satisfies*

$$1 - \frac{1}{rN} \sum_{i \in I_N} \xi_i \leq \frac{\mathbb{E}[N_s]}{N} \leq \frac{1}{N} \sum_{i \in I_N} \exp\!\left[-\frac{(r-\xi_i)^2}{16M^2}\right], \tag{11}$$

*where*

$$\xi_i^2 \;=\; M^2 \sum_{\substack{j \in I_N \\ j \neq i}} \alpha_j^2 \;+\; \left(M^2 - \tfrac{\delta^2}{2}\right) \sum_{\substack{j,k \in I_N \\ j \neq k \neq i}} \alpha_j \alpha_k. \tag{12}$$

The quantity $\xi_i$ measures how widely the other $N-1$ selected embeddings, rescaled by their weights, are spread around $x_i$. If $\xi_i$ is small, most selected points cluster near the context vector $s$, so many of them fall inside the ball $B_r(s)$ and become distinguishable. The lower bound in (11) subtracts from one a penalty proportional to the cumulative spread $\sum_i \xi_i$; the tighter the cluster (smaller $\xi_i$), the closer the ratio $\mathbb{E}[N_s]/N$ is to 1. Conversely, the upper bound shows that once $r$ is smaller than $\xi_i$ for many $i$, the exponential term decays rapidly, implying very few embeddings remain separable under the chosen radius.

## 2.3 Gradient Sensitivity of Attention

The results above show that a language model must sharply distinguish informative from non-informative tokens; in other words, the attention weight distribution should be as selective as possible. Refining that distribution, however, exposes a second difficulty: *gradient sensitivity* during training.

Consider two nearly identical logit vectors

$$\boldsymbol{\ell}^{(1)} = (0, \ldots, 0, \, a, \, a + \varepsilon), \qquad \boldsymbol{\ell}^{(2)} = (0, \ldots, 0, \, a + 2\varepsilon, \, a),$$

so that their Euclidean distance satisfies $\|\boldsymbol{\ell}^{(1)} - \boldsymbol{\ell}^{(2)}\|_2 = \sqrt{5}\,\varepsilon$. Let $\boldsymbol{\alpha}^{(1)}, \boldsymbol{\alpha}^{(2)}$ be the corresponding softmax weights (Equation (2)) and denote by $\nabla_\ell \boldsymbol{\alpha}$ the Jacobian of the softmax map. A first-order expansion gives

$$\left\|\boldsymbol{\alpha}^{(1)} - \boldsymbol{\alpha}^{(2)}\right\|_2 \;\approx\; \left\|\nabla_\ell \boldsymbol{\alpha}^{(1)} \, (\boldsymbol{\ell}^{(1)} - \boldsymbol{\ell}^{(2)})\right\|_2 \;\sim\; \sqrt{2}\,\frac{\varepsilon}{T},$$

because the largest two logits swap order and the associated softmax gradient scales as $1/T$. Hence, even though the logits differ by only $O(\varepsilon)$, the output distribution can change by $O(\varepsilon/T)$. For a sufficiently small temperature $T$ (a common tactic for sharpening attention) this factor becomes large, making the gradient step volatile and potentially destabilising optimisation.

This example highlights a fundamental trade-off: stronger normalisation (smaller $T$, or any variant that steepens the softmax) improves token separability but simultaneously amplifies gradient variance, complicating training of deep transformers.

Let have a closer look at how normalisation affects gradient descent. For the general scheme in (3), let $\{l_i\}_{i=1}^L$ be the logits and $\{\alpha_i\}_{i=1}^L$ the resulting attention weights. We characterise the magnitude of the Jacobian $\nabla_l \boldsymbol{\alpha}$.

**Lemma 2.** *With the notation above,*

$$\|\nabla_l \boldsymbol{\alpha}\|_2 \leq \min\left\{ \|F'\|_2 \left( \frac{1}{L \min_j F(l_j, \theta)} + \frac{\|F\|_2}{L^2 \min_j F^2(l_j, \theta)} \right), \sqrt{2} \right\} \tag{13}$$

*where $\|F\|_2 = \max_i |F(l_i, \theta)|$ and $\|F'\|_2 = \max_i |F'(l_i, \theta)|$.*

The upper bound grows when any logit $l_j$ pushes $F(l_j, \theta)$ toward zero, because both terms inside the parentheses scale inversely with $\min_j F(l_j, \theta)$. Thus, a sharply peaked normalisation—softmax with a small temperature, for instance—induces large Jacobian norms, signalling high gradient sensitivity during training.

**Corollary 3.** *For the softmax normalisation in (2),*

$$\left\|\nabla_l \boldsymbol{\alpha}\right\|_2 \;\leq\; \min\left\{\frac{1}{4T}, \, \sqrt{2}\right\}.$$

A smaller temperature $T$ sharpens the softmax but simultaneously inflates the Jacobian norm (scaling as $1/T$), making the attention distribution highly sensitive to even tiny logit perturbations and thus harder to train stably.

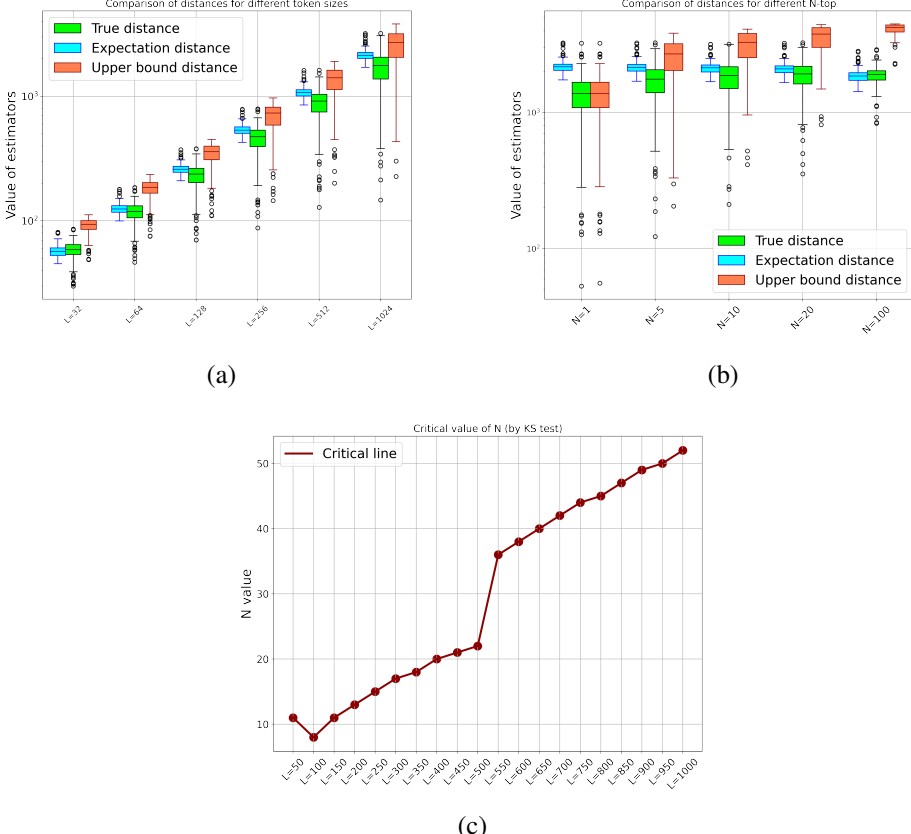

<p align="center">(a)</p>
<p align="center">(b)</p>
<p align="center">(c)</p>

Figure 2: **Distance statistics validate Theorem 1.** (a) With $N = 5$, both the true distance (green) and its expectation (blue) grow roughly linearly in $L$; the red upper bound is safe but conservative. (b) With $L = 1024$, increasing $N$ beyond 20 yields diminishing returns: the distance plateaus while the bound tightens. (c) Critical top-$N$ obtained by a KS test ($\alpha = 0.01$); fewer than 6 % of the tokens need to be selected before the empirical and expected distances become statistically indistinguishable.

## 3 Experiments

We evaluate our theoretical findings on the publicly available GPT-2 model family[1] [15]. All text is tokenised with byte-pair encoding (BPE) [6] as implemented in the Hugging Face `transformers` library. Unless otherwise stated, the input consists of consecutive chapters from *War and Peace* by Leo Tolstoy (public domain), providing long-form prose well beyond the model's context window. For every layer and attention head, we extract the full attention matrix $A \in \mathbb{R}^{L \times L}$ and the associated query, key, and value tensors, enabling direct comparison with our distance and geometry metrics. Implementation details, hyperparameters, and reproducibility scripts are included in Appendix B.

### 3.1 Distance analysis

We now test the non-asymptotic bounds of Theorem 1. Implementation details appear in Appendix B. Two complementary experiments are performed:

1. *Scaling with sequence length.* Fix $N = 5$ and vary $L \in \{32, \dots, 1024\}$.
2. *Scaling with top-$N$.* Fix $L = 1024$ and vary $N \in \{1, 5, 10, 20, 100\}$.

---

[1]We report results for the 124 M parameter version; qualitatively identical trends were observed for larger variants.

For each configuration we compute, across all 144 GPT-2 heads/layers, (i) the true distance $\tilde{d}$ (7), (ii) the expectation term of Theorem 1, and (iii) the analytic upper bound. In addition, we estimate a *critical* top-$N$ value: the smallest $N$ for which the empirical and expected distance distributions are indistinguishable under a two-sample Kolmogorov–Smirnov test ($\alpha = 0.01$).

Results summarised in Fig. 2 support following key observations. **(i)** For $N \ll L$ the distance scales linearly with sequence length, exactly as Corollary 2(i) predicts. **(ii)** As $N$ approaches 100, the true and expected distances converge while the upper bound tightens—evidence for the $N \to L$ collapse of Eq. (9). **(iii)** The critical-$N$ curve grows sub-linearly ($\approx 0.06L$), confirming that only a small subset of tokens can be separated before attention behaves as if weights were uniform.

These empirical trends corroborate the theoretical claim that softmax normalisation retains discriminative power only when the active set is a small fraction of the context; larger $N$ values chiefly add noise.

## 3.2 Geometric Separability

We now quantify how many of the $N$ highest-weight tokens remain *geometrically distinguishable* according to Theorem 2. For each sequence we set $r = \min\limits_{i \notin I_N} \|s - \alpha_i x_i\|_2$, so that every non-selected token lies outside the ball $B_r(s)$. Using GPT-2 embeddings normalised as in Assumptions 1–2, we compute the empirical ratio $N_s/N$ (Definition (10)) across all heads and layers and compare it with the analytic bounds of Theorem 2; results are summarised in Fig. 3.

Across the full model, the proportion of distinguishable tokens declines up to $N \approx 16$ and then saturates between $0.7$ and $0.85$. The exponential *upper* bound tracks the empirical maxima closely, showing the theorem is tight on the high side, whereas the *lower* bound is intentionally conservative. Thus—even with idealised spherical embeddings—softmax attention cannot cleanly separate more than about four-fifths of the tokens it selects; adding further tokens mainly dilutes the representation without improving geometric resolution.

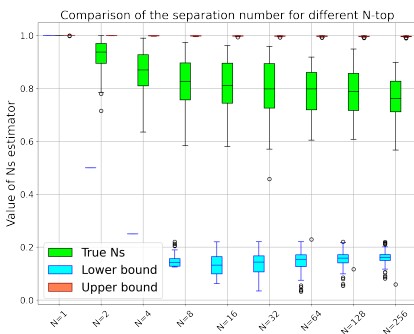

Figure 3: **Geometric separability saturates at 70–85%.** For increasing top-$N$, the empirical fraction of distinguishable embeddings $N_s/N$ (green boxes) quickly plateaus; roughly one-fifth of selected tokens remain outside $B_r(s)$.

## 3.3 Gradient sensitivity

We next explore how the softmax temperature $T$ and logit perturbation scale $\varepsilon$ affect the stability of the attention map. For each head–layer pair we evaluate the finite-difference Jacobian norm

$$g(T,\varepsilon) = \frac{1}{\varepsilon}\left\|\boldsymbol{\alpha}_{\boldsymbol{\ell}+\varepsilon\Delta\boldsymbol{\ell}} - \boldsymbol{\alpha}_{\boldsymbol{\ell}}\right\|_2, \qquad \|\Delta\boldsymbol{\ell}\|_2 = 1,$$

which approximates $\|\nabla_{\boldsymbol{\ell}}\boldsymbol{\alpha}\|_2$. Full implementation details are provided in Appendix B. Figure 4 shows the *maximum* value of $g(T,\varepsilon)$ across all 144 heads/layers of GPT-2 for $\varepsilon \in \{10^{-3}, 10^{-1}, 10\}$.

For $T < 0.1$ the empirical curves follow the theoretical $1/T$ trend of Corollary 3; smaller $\varepsilon$ values yield larger gradients because the sharpest logits dominate the variation. Once $T \geq 1$ all curves collapse and drop by two orders of magnitude, indicating much improved robustness to logit perturbations but a concomitant loss of selectivity. These findings confirm the trade-off already highlighted by our theory: *sharper softmax improves token separability yet*

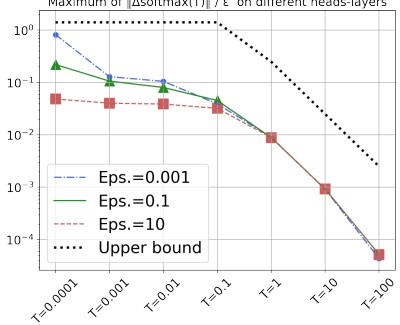

Figure 4: **Gradient sensitivity decays as $1/T$.** Maximum finite-difference Jacobian norm $g(T,\varepsilon)$ for three perturbation magnitudes (coloured curves, log–log scale). The dashed black curve is the theoretical bound $\min\{1/(4T), \sqrt{2}\}$ from Corollary 3.

*inflates gradient variance*, whereas higher temperatures stabilise training at the cost of blurrier attention.

## 4  Discussion

Softmax remains the default normalisation in modern Transformers because it is simple, differentiable and lends a probabilistic interpretation to the weights [18]. Its limitations, however, are now well established: it produces dense distributions, its peak value drops as the context grows (*vanishing–attention* effect), and its Jacobian explodes when the temperature is driven low, yielding unstable gradients. Three main research lines have emerged to address these issues.

1. *Sparsity-inducing rules* such as Sparsemax and $\alpha$–Entmax replace the softmax exponential with projections onto a simplex, producing exact zeros and lowering entropy [12].

2. *Length-aware rescales* such as Scalable–Softmax scale logits by $\log L$ so that the maximum attention weight stays roughly constant as the sequence grows, alleviating the vanishing–attention problem [13].

3. *Gradient-controlled variants* (e.g. Self-Adjusted Softmax) re-weight logits according to the layer-wise dynamic range, keeping the Jacobian spectrum in a healthy band [22].

Our work complements these efforts by *explaining, in closed form, why the above modifications are necessary*. Where previous papers introduced a new rule and showed empirical gains, we derived non-asymptotic bounds (Theorems 1–2) and gradient limits (Lemma 2) that *apply to any normalisation function $F(\cdot, \theta)$*. In particular:

- The **distance bound** shows that when $N$ grows proportionally to $L$, the representation distance $\tilde{d}$ necessarily collapses to zero, formalising the empirical softmax bottleneck noted by [20]. Our GPT-2 experiment (Fig. 2, middle) confirms the predicted plateau and justifies why sparse or length-aware rules can improve long-context performance.

- The **geometric bound** states that even under optimistic spherical assumptions no more than $\approx 80\%$ of the top-$N$ tokens can stay inside the selective ball (Fig. 3). This explains why empirical studies report that multiple heads are required to cover distinct parts of the context: a single head cannot make every "important" token simultaneously salient. The reason behind this idea is pretty simple. Assuming the independence of the heads as attention mechanism parts, we can conclude that for given separability level $p$, $H$ heads can cover up to $1 - (1 - p)^H$ of the top-N tokens. Under our analysis, we see that for $p = 0.8$ it is needed approximatelly $H = 3$ to cover up the 99% of information.

- The **gradient bound** recovers the familiar $1/(4T)$ law but also shows how any normaliser with a vanishing minimum value will inherit the same instability. Figure 4 demonstrates that GPT-2 operates close to the theoretical limit when $T < 10^{-1}$, validating the usefulness of gradient-controlled variants such as SA-Softmax.

The combined theory–experiment picture suggests three practical guidelines.

1. **Keep the active set small.** The critical-$N$ curve in Fig. 2 grows roughly like $0.06L$; selecting more tokens yields vanishing returns and erodes separability. Top-$k$ or sparse attention should be preferred when $L \gg k$.

2. **Monitor attention entropy.** A rising entropy or a drop in the empirical $N_s/N$ ratio is an early sign that a head has saturated its geometric capacity; adding additional heads or switching to a length-aware normaliser can restore separability.

3. **Avoid overly sharp softmax.** Lowering $T$ below $10^{-1}$ increases Jacobian norms without increasing separability (Figs. 3, 4). Practitioners should instead use normalisers that decouple selectivity from gradient health (e.g. Sparsemax, SS-Max, or SA-Softmax).

Our analysis assumes embeddings are *a priori* L2-normalised and roughly isotropic. Real models may violate these assumptions, and future work should extend the geometric bound to non-spherical distributions. Another promising direction is to design a length-adaptive, gradient-controlled normaliser that inherits the best properties of Sparsemax (sparsity), SS-Max (length awareness) and SA-Softmax (stable Jacobians) while admitting proofs analogous to Theorems 1–2.

# 5 Conclusions

This work provides theoretical and empirical analysis of normalisation in attention mechanisms beyond the classical softmax. We derived two non-asymptotic bounds (Theorems 1 and 2) that link token separability to sequence length $L$, selection size $N$, and the embedding geometry, and we established a general Jacobian bound (Lemma 2) that explains the well-known trade-off between sharpness and gradient stability. Experiments on GPT-2 confirmed all three predictions:

- the representation distance collapses once $N$ grows proportionally to $L$;
- no more than $\approx 80\%$ of the selected tokens can be geometrically distinguished, even under ideal spherical embeddings;
- the empirical Jacobian norm tracks the theoretical $1/(4T)$ law, saturating at low temperature.

Taken together, these results recast softmax attention as a *selective but capacity-limited aggregator*: it discriminates well only while the active set is a small fraction of the context. The analysis also clarifies why recently proposed normalisers—Sparsemax, Scalable-Softmax, and Self-Adjusted Softmax—offer complementary benefits: they relax one or more of the intrinsic limits quantified here.

**Practical takeaways.**    (1) limit the top-$k$ set to a sub-linear function of the context length; (2) monitor attention entropy or the $N_s/N$ ratio during training; and (3) prefer length-aware or sparsity-inducing normalisers over aggressive temperature scaling.

Overall, our experiments shows several limitations providing the practical diagnostic idea:

1. When the level of geometrical separability drops to $70 - 80\%$, this signifies that the head has saturated its geometric capacity.
2. The temperature limitations shows that making the distribution sharp makes the Jacobian norm exploding. The teoretical and practical analysis shows that it's better to avoid using the $T \leq 0.1$.

**Future directions.**    Immediate extensions include (i) relaxing the spherical-embedding assumption, (ii) analysing multi-query and multi-head interactions, and (iii) designing a single normalisation rule that is simultaneously length-adaptive, sparse, and gradient-stable. We hope the quantitative framework introduced here will serve as a benchmark for such developments and, more broadly, for principled improvements to long- context transformers.

Overall, the present study provides theoretical footing for the growing body of work that modifies softmax, and furnishes quantitative tools for diagnosing when a given attention head has reached its intrinsic limit.

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

# A  Theory justification

In the given appendix, we provide the proofs for our theory part.

*Lemma 1.* Consider the $\alpha_i$ term:

$$\alpha_i = \frac{F(l_i, \theta)}{\sum_{j=1}^{L} F(l_j, \theta)} \leq \frac{C_2'(\theta)}{\sum_{j=1}^{L} C_1'(\theta)} \leq \frac{C_2(\theta)}{L}, \tag{14}$$

since the function is continuous on a compact space. The lower bound works the same.

For the proof of the corollary, notice that $l_i = \mathbf{q}^\top \mathbf{k}_i$, therefore $|l_i|$ is bounded by norm of $\|\mathbf{q}\|\|\mathbf{k}\|$. Hence, we have the following bound. $\square$

*Theorem 1.* 1. Here we have fixed $I_N = \{i_1, \ldots, i_N\} \subset [1, \ldots, L]$. Therefore, we have:

$$\tilde{d} = \sum_{i \in I \backslash I_N} \left\| \alpha_i x_i - \sum_{j \in I_N} \alpha_j x_j \right\| \leq \sum_{i \in I \backslash I_N} \sum_{j \in I_N} \alpha_j \left\| \frac{\alpha_i}{\bar{\alpha}_N} x_i - x_j \right\| = \tag{15}$$

$$\sum_{i \in I \backslash I_N} \sum_{j \in I_N} \alpha_j \left\| \frac{\alpha_i}{\bar{\alpha}_N} x_i - \frac{\alpha_i}{\bar{\alpha}_N} x_j + \frac{\alpha_i}{\bar{\alpha}_N} x_j - x_j \right\| \leq \tag{16}$$

$$\sum_{i \in I \backslash I_N} \sum_{j \in I_N} \alpha_j \left( \frac{\alpha_i}{\bar{\alpha}_N} d_1 + \|x_j\| \left| 1 - \frac{\alpha_i}{\bar{\alpha}_N} \right| \right) = \tag{17}$$

$$(1 - \bar{\alpha}_N) d_1 + \max_{j \in I_N} \|x_j\| \left[ \bar{\alpha}_N (L - N) - (1 - \bar{\alpha}_N) \right], \tag{18}$$

where $d_1 = \max_{i \notin I_N, j \in I_N} \|x_i - x_j\|$.

2. For the probability part:

$$E = \mathbb{E}\left[ \sum_{i \notin I_N} \|\alpha_i x_i - s\| \right] = \mathbb{E}\left[ \sum_{i=1}^{L} \mathbf{1}\left(i \notin I_N\right) \|\alpha_i x_i - s\| \right] \approx \tag{19}$$

$$\sum_{i=1}^{L} \mathbb{P}(i \notin I_N) \mathbb{E}\left[ \|\alpha_i x_i - s\| | i \notin I_N \right]. \tag{20}$$

We can estimate both the expectation of norm and probability terms as follows:

$$\mathbb{P}(i \notin I_N) = \frac{L - N}{N}, \tag{21}$$

$$\mathbb{E}\left[ \|\alpha_i x_i - s\| | i \notin I_N \right] = \mathbb{E}\left[ \left\| \alpha_i x_i - \sum_{j \neq i} \mathbf{1}(j \in I_N) \alpha_j x_j \right\| \, \bigg| \, i \notin I_N \right] \approx \tag{22}$$

$$\left\| \alpha_i x_i - \frac{N}{L - 1} \sum_{j \neq i} \alpha_j x_j \right\|. \tag{23}$$

As a result, we obtain:

$$E = \frac{L - N}{L} \sum_{i=1}^{L} \left\| \alpha_i \left( 1 + \frac{N}{L - 1} \right) x_i - \frac{N}{L - 1} \bar{x} \right\| + \varepsilon \tag{24}$$

The error term between the approximation and the true value can be estimated using Jensen's gap bound:

$$\varepsilon_i = \mathbb{E}[\|\alpha_i x_i - s\| | i \notin I_N] - \|\alpha_i x_i - \mathbb{E}[s | i \notin I_N]\| \leq \frac{1}{2} \frac{N(L-N-1) \sum_{j \neq i} \alpha_j^2 \|x_j\|^2}{\|\alpha_i x_i - s_i\|} \quad (25)$$

Therefore:

$$\varepsilon \leq \sum_{i=1}^{L} \frac{L-N}{L} \varepsilon_i \quad (26)$$

Now let's move to the corollary section.

1. Assuming $N \ll L$ and $L$ grows, we have:

$$E \approx \sum_{i=1}^{L} \alpha_i \|x_i\| \quad (27)$$

2. When $N \to L$, we have $E \to 0$, since number of outer elements goes to zero.

$\square$

*Theorem 2.* We have:

$$\mathbb{E}[N_s] = \sum_{i \in I_N} 1(\|s - \alpha_i x_i\| \leq r) = \sum_{i \in I_N} \mathbb{P}(\|s - \alpha_i x_i\| \leq r) \quad (28)$$

Hence, we need to estimate the probability of the $\alpha_i x_i$ being in the sphere.

Notice that $\|\alpha_i x_i - s\|$ is bounded random variable. We can estimate it as $\|\alpha_i x_i - s\| \in [0, 2M]$. Hence, we have a Hoeffding-type inequality:

$$\mathbb{P}(X_i \leq r) \leq \inf_t [e^{-rt} \mathbb{E} e^{X_i t}] \leq \exp\left[\inf_t \left(-rt + t\mathbb{E}X_i + 4M^2 t^2\right)\right] \leq \exp\left[-\frac{(r - \mathbb{E}X_i)^2}{16M^2}\right], \quad (29)$$

where expected value of $X_i = \|s - \alpha_i x_i\|$ can be estimated as follows:

$$\mathbb{E}X_i \leq \sqrt{\mathbb{E}X_i^2}, \quad (30)$$

and for the squared norm, we have:

$$\|X_i\|^2 = \left\|\alpha_i x_i - \sum_{j \in I_N} \alpha_j x_j\right\|^2 = \left\|\sum_{j \in I_N, j \neq i} \alpha_j x_j\right\|^2 = \sum_{\substack{j,k \in I_N \\ j \neq i \\ k \neq i}} \alpha_j \alpha_k \langle x_i, x_j \rangle \leq \quad (31)$$

$$M^2 \sum_{\substack{j \in I_N \\ j \neq i}} \alpha_j^2 + \left(M^2 - \frac{\delta^2}{2}\right) \sum_{\substack{j,k \in I_N \\ j \neq k \neq i}} \alpha_j \alpha_k \equiv \xi_i^2, \quad (32)$$

where the last bound caused by condition $\|x_i - x_j\|^2 \geq \delta^2$.

Therefore, we have:

$$\mathbb{E}[N_s] \leq \sum_{i \in I_N} \exp\left[-\frac{(r - \xi_i)^2}{16M^2}\right] \tag{33}$$

The lower bound is easier. Using Markov inequality and Cauchy-Schwarz (30):

$$\mathbb{P}(X_i \leq r) = 1 - \mathbb{P}(X_i > r) \leq 1 - \frac{\mathbb{E}X_i}{r} \geq 1 - \frac{\xi_i}{r} \tag{34}$$

Hence, we have:

$$N - \frac{1}{r}\sum_{i \in I_N} \xi_i \leq \mathbb{E}[N_s] \leq \sum_{i \in I_N} \exp\left[-\frac{(r - \xi_i)^2}{16M^2}\right] \tag{35}$$

$\square$

# B   Experiments

In the appendix, we provide details of the whole experiment setup and give the pseudocode we implemented for each figure.

**System parameters**

For the given research, we used the Apple M1 Pro chip with a 10-core CPU and 16GB of unified memory, based on ARM architecture.

**Software framework**

The models were implemented and examined using PyTorch [14], running on the Apple M1 Pro's ARM-based CPU architecture to ensure efficient computation.

For the parallelization procedure, we used Joblib library [9].

**Distance analysis**

---
**Algorithm 1** Distance Analysis. Different $L$ and fixed $N$.

---
**Require:** Text input, list of token lengths $L_{\text{values}}$, fixed $N = 5$
**Ensure:** Averaged distance statistics across layers and heads
1:  **for** $L$ in $L_{\text{values}}$ **do**
2:      Encode text using GPT-2 and extract attention matrices for all heads and layers
3:      **for** each (head, layer) in GPT-2 **do**
4:          **for** each token index $t$ in $1, \ldots, L$ **do**
5:              Compute true distance $\tilde{d}$ via Eq. (7)
6:              Compute upper bound $d_{\max}$ via Eq. (9)
7:              Compute expectation $E$ via Eq. (10)
8:              Store $(\tilde{d}, d_{\max}, E)$
9:          **end for**
10:         Average distances across tokens
11:         Store result for (head, layer)
12:     **end for**
13:     Store all results for current $L$
14: **end for**
15: **return** All distance statistics

---

**Algorithm 2** Distance Analysis. Different top-N and fixed $L$

**Require:** Text input, list of top-$N$ values $N_{\text{values}}$, fixed $L = 1024$
**Ensure:** Averaged distance statistics across layers and heads
1: Encode text using GPT-2 and extract attention matrices for all heads and layers
2: **for** $N$ in $N_{\text{values}}$ **do**
3:     **for** each (head, layer) in GPT-2 **do**
4:         **for** each token index $t$ in $1, \ldots, L$ **do**
5:             Compute true distance $\tilde{d}$ via Eq. (7)
6:             Compute upper bound $d_{\max}$ via Eq. (9)
7:             Compute expectation $E$ via Eq. (10)
8:             Store $(\tilde{d}, d_{\max}, E)$
9:         **end for**
10:         Average distances across tokens
11:         Store result for (head, layer)
12:     **end for**
13:     Store all results for current $N$
14: **end for**
15: **return** All distance statistics

---

**Algorithm 3** Distance Analysis. Critical Top-$N$ detection.

**Require:** Text input, $L$, top-$N$ values $N_{\text{values}}$, significance level $\alpha = 0.05$
**Ensure:** First $N$ for which expected and true distances are statistically close
1: Encode text using GPT-2 and extract attention matrices
2: **for** $N$ in $N_{\text{values}}$ **do**
3:     Initialize list of relative errors
4:     **for** each (head, layer) in GPT-2 **do**
5:         **for** each token $t$ in $1, \ldots, L$ **do**
6:             Compute true distance $\tilde{d}$ via Eq. (7)
7:             Compute expected distance $E$ via Eq. (10)
8:         **end for**
9:         Compute mean true distance $\bar{d}$
10:         Compute mean expected distance $\bar{E}$
11:         Store the $\bar{d}$ and $\bar{E}$
12:     **end for**
13:     **if** Kolmogorov-Smirnov($\bar{d}, \bar{E}$, significance level $\alpha$) is true **then**
14:         **return** $N$
15:     **end if**
16: **end for**
17: **return** $-1$                                   ▷ No $N$ meets the condition

**Geometrical analysis**

---

**Algorithm 4** Geometrical analysis. Separation Ratio and Bounds for Top-N Attention Tokens.

---

**Require:** Text input, sequence length $L$, top-$N$ values $N_{\text{values}}$
**Ensure:** Box plots of $N_s/N$ and its theoretical bounds
 1: Encode text using GPT-2 and extract attention matrices
 2: Extract and normalize token embeddings
 3: **for** $N$ in $N_{\text{values}}$ **do**
 4:     **for** each (head, layer) in GPT-2 **do**
 5:         **for** each token $t$ in $1, \ldots, L$ **do**
 6:             Compute $N_s/N$ via direct counting
 7:             Compute lower and upper bounds from Theorem 2
 8:         **end for**
 9:         Average $N_s/N$, upper bound, and lower bound over all tokens
10:         Store results for (head, layer)
11:     **end for**
12:     Store aggregated results for $N$
13: **end for**
14: Generate box plots comparing true and theoretical values

---

**Gradient analysis**

---

**Algorithm 5** Gradient Sensitivity Analysis

---

**Require:** Text, temperature values $T_{\text{values}}$, shift values $\varepsilon_{\text{values}}$
**Ensure:** Sensitivity statistics across temperatures and shifts
 1: Convert text to logits matrices for all (head, layer) pairs
 2: **for** $T$ in $T_{\text{values}}$ **do**
 3:     **for** $\varepsilon$ in $\varepsilon_{\text{values}}$ **do**
 4:         **for** each (head, layer) **do**
 5:             **for** each token $t \in \{1, \ldots, L\}$ **do**
 6:                 Sample unit vector $v$ with $\|v\|_2 = 1$
 7:                 Compute shifted logits $l' = l + \varepsilon v$
 8:                 Compute softmax distributions $\alpha = \text{softmax}(l/T)$, $\alpha' = \text{softmax}(l'/T)$
 9:                 Compute sensitivity: $\delta = \|\alpha' - \alpha\|_2/\varepsilon$
10:             **end for**
11:             Average $\delta$ over tokens
12:         **end for**
13:         Store maximum average $\delta$ across (head, layer) for current $\varepsilon$
14:     **end for**
15:     Store results for temperature $T$
16: **end for**
17: **return** Sensitivity statistics

---

**Time resources**

Here, we provide the time execution for all algorithms:

| Algorithm | Time of execution | Parallelization |
|-----------|-------------------|-----------------|
| Alg.1 | 24 min. | No |
| Alg.2 | 37 min. | No |
| Alg.3 | 17 h. 4min. | Yes |
| Alg.4 | 7 min. | No |
| Alg.5 | 1 min. | No |

Table 1: Comparison of algorithm execution times. The table references algorithms defined in Algorithms 1–5, highlighting their respective performance durations and whether they used parallelization.

