# OpenReview forum: "Limitations of Normalization in Attention"
_NeurIPS.cc/2025/Conference — NeurIPS 2025 poster_

### Official Review · Reviewer_NWWE · 2025-06-08

**Clarity:** 2
**Significance:** 2
**Originality:** 2
**Rating:** 4
**Confidence:** 3

**Summary:**

This paper revisits the token selection mechanism with the softmax function, which is currently a core component of the attention mechanism of the transformer architectures.
The authors begin by analyzing how, as the context length $L$ increases, the attention mechanism becomes less capable of efficiently picking the relevant tokens.
They further investigate the geometric token separability and the gradient instability when the softmax temperature $T$ decreases.
The paper also includes the experiments using the pre-trained GPT-2 models to support the theoretical findings.

**Questions:**

1. Line 59: Does the function $F$ takes values only in the positive real domain, rather than the entire $\mathbb{R}$? I assume the constants $C_1^\prime(\theta)$ and $C_2^\prime (\theta)$ are positive in the proof of the lemma 1, but the paper lacks clarification.
2. I'm struggling to understand the significance of the analysis in Section 2.2. How does this result relate to the standard softmax normalization, for example, in terms of the number of selected examples $N$? The finding that a larger $N$ leads to a smaller ratio $N_s / N$ (in Figure 3) seems unsurprising. This appears to be a general phenomenon in token separation, rather than a specific limitation of the softmax.

Additionally, I would appreciate a clarification and resolution of the concerns raised in the Weaknesses section.

**Ethical Concerns:**

["NO or VERY MINOR ethics concerns only"]

**Final Justification:**

Initially, I found it challenging to understand the motivation of this work, its positioning within the existing literature, and the motivations and connections across the different analyses and sections, which led me to assign a low score. However, the rebuttal has resolved my questions. Since the authors have promised to improve these aspects in the revision, I now recommend acceptance.

**Limitations:**

yes

**Paper Formatting Concerns:**

No issues.

**Quality:**

2

**Strengths And Weaknesses:**

**Strength**
1. The paper approaches the limitation of the softmax from three distinct perspectives, each of which is explained well.
2. The discussion and conclusion provide the practical takeaways. It helps clarify the overall message of the paper.
3. The inclusion of pseudo-codes improves the reproducibility of the empirical results.

**Weakness**
1. The paper lacks a strong foundational motivation. It is unclear why the authors focus on the limitation of softmax-based token selection, given its widespread success in many machine learning applications.
2. In equation 5, the probability of token assignment becomes $O(1/L)$, but it is not obvious whether it leads to vanishing attention in practice. This issue becomes problematic when the sequence length $L$ grows exponentially with respect to $\Delta = \\|\boldsymbol{\mathbf{q}}\\|_2 \\|\boldsymbol{\mathbf{k}}\\|_2$, which seems unrealistic in the practical scenarios. As a result, this point does not convincingly provide the weakness of the softmax normalization. At least, the authors should demonstrate that such uniform behavior actually occurs in some realistic setting and that it leads to degraded performance, either through new experiments or by citing existing work.
3. The paper lacks "Related Work" section. It obscures the position of the paper and makes it difficult to assess the novelty of the contribution.
4. Several alternatives to softmax are presented in "Discussion" section, but it is unclear whether they actually address the issues of softmax normalization and lead to the performance improvement. I believe that the experimental results comparing the softmax with them would enhance the quality of the paper.

---

> ### Author Rebuttal · Authors · 2025-07-28
>
> We thank the reviewer for their detailed and in-depth feedback. We address each point below:
>
> *Weaknesses:*
>
> **W1.** We thank the reviewer for this comment and agree that a clearer statement of foundational motivation strengthens the paper.  Although softmax‐based attention has achieved impressive empirical success in many settings, modern applications increasingly demand handling very long contexts (tens of thousands of tokens) and fine‐grained focus on rare but critical information.  In these regimes, practitioners have observed two persistent problems:
>
>  1. **Vanishing‐attention in long contexts.**  As $L$ grows, softmax weights collapse toward $1/L$, making it impossible to single out informative tokens (flagship models can effectively use about 10-15% of their context window as demonstrated by BABILong benchmark) — a phenomenon that underlies many recent “long‐context” workarounds (sparse or windowed attention, segment level recurrence, compressive memories, etc.).
>
>  2. **Gradient instability under sharpening.**  To counteract vanishing attention, one often lowers the temperature $T$, but this in turn inflates the softmax Jacobian ($\sim1/(4T)$), leading to noisy or divergent training dynamics.
>
> There has been no quantitative, non‐asymptotic theory explaining why and when these failures occur, nor any general tools to predict the limits of a given attention rule.  Our work fills this gap by:
>
>
>  1. Deriving tight bounds that show exactly how the context‐vector vs. non‐selected‐token distance collapses once the selected set grows proportionally with $L$.
>  2. Proving a fundamental geometric ceiling on the fraction of top‐$N$ tokens a single head can distinguish, even in idealized spherical embeddings.
>  3. Quantifying the precise $O(1/T)$ trade‐off between selectivity and gradient‐sensitivity for any smooth normalizer.
>
> These results provide the first unified theoretical framework to explain the intrinsic capacity limits of softmax attention and to guide the design of more robust normalization schemes—an essential foundation for scaling Transformers to ever‐longer contexts without sacrificing stability or accuracy.
>
> We will extend the Introduction to better present motivation by clearly stating these points and clarifying why understanding softmax’s limitations is critical despite its empirical success in shorter‐context regimes.
>
> **W2.** We thank the reviewer for raising this point.  Although Equation 5 shows that each attention weight scales as $O(1/L)$, in practice, modern transformers keep representations normalized, thus bounding the norm. Another factor is that $\Delta = ||q||_2 ||k||_2 $ remains constant when $L$ varies from $10^2$ to $10^6$. Consequently, when $L$ reaches sizes $L\sim10^5$, the softmax denominator far exceeds any fixed numerator, driving maximum probability $\alpha \approx Const/L$ and $Const \ll L$.  In other words, even the most relevant token receives a vanishingly small probability.
> A concise exploration of this phenomenon – a fix via explicit length‐aware scaling—is given in Nakanishi et al. (2025) “Scalable‑Softmax Is Superior for Attention”  [12], which we highlight in Section 5.  By adding a $\log L$ term to the logits before exponentiation, Scalable‑Softmax keeps $\max_i \alpha_i$ roughly constant as $L$ grows, thereby preserving focus on the most informative tokens even in very long contexts.
>
> **W3.** We thank the reviewer for this suggestion. To clarify the paper’s position and highlight its novel contributions, we will extend the paper to explicitly contrast our theoretical framework with existing qualitative or single‐aspect treatments in sufficient detail. We believe this contextualization will strengthen the reader’s ability to assess the novelty and impact of our contributions.
>
> **W4.** We thank the reviewer for this suggestion. Our primary aim in the present work was to develop a general, closed‐form theoretical framework for understanding the intrinsic limits of any softmax‐type normalizer, and to validate those limits empirically on a standard Transformer backbone.  As such, we have not yet carried out a systematic, head‐to‐head empirical comparison between softmax and its sparse or length‐aware variants (Sparsemax, Scalable‐Softmax, Self‐Adjusted Softmax, etc.).
>
> Because our bounds apply out of the box to any differentiable normalizer $F$, we can now conduct such comparisons in an “informed” manner: for each candidate $F$ one simply computes its distance‐collapse curve, geometric ceiling, and gradient‐sensitivity estimate under our framework, and then measures actual performance on long‐context tasks. We view this as the natural next step—and indeed a full paper in its own right—where we will evaluate a range of alternative attention rules on LLaMA, Qwen and other modern LLMs, identify which normalization choices yield the best trade‐offs in practice, and use our bounds to explain why they succeed or fail.   We believe that separating the present foundational treatment from a dedicated large‐scale empirical evaluation yields a clearer exposition and more coherent research agenda.
>
> *Questions:*
>
> **Q1.** Yes, the function $F$ is assumed to be strictly positive over a compact domain (otherwise the Extreme value theorem won’t work); we will clarify this explicitly in Lemma 1. Thank you for spotting this!
>
> **Q2.** We agree that increasing $N$ generally reduces $N_s/N$, but our analysis quantifies how quickly this degradation occurs under softmax. The key insight is that separability saturates for $N \ll L$, supporting the claim that softmax attention is inherently capacity-limited.
>
> We hope these changes address your concerns and improve the clarity and significance of our contribution.
>
> Thank you for your questions and the detailed review.

---

> > ### Comment · Reviewer_NWWE · 2025-08-03
> >
> > Thank you for your detailed reply. Since I now feel satisfied with the motivation of this work, I will update the score to 3. The discussion in Section 2.2, which I had not fully appreciated at first, now appears significantly more interesting after rereading it along with your explanation. I also appreciate your plan to add the related work section, which I believe will further clarify the position of the paper.
> >
> > I still have some unresolved questions, particularly regarding Corollary 2 and the Figure 2. My current understanding is as follows, if possible, I would appreciate your confirmation or comments:
> > 1. The main goal here is to evaluate whether the top-N token selection leads to meaningful selection, by comparing the actual distance $\tilde{d}$ with the expected distance $E$ from the totally random selection. This analyzes the quality of token selection in terms of the *distance* of token separability.
> > 2. However, since this analysis focuses purely on distance, it does not account for the utility of the *direction* of the selected tokens. This is instead addressed in the geometric analysis in Section 2.2.
> >
> > Regarding the results, I believe that I now understand the message behind the diminishing returns observed in Figure 2 (middle) as $N$ increases, as well as the limit around $\approx 0.06L$ in Figure 2 (right), where the empirical distance becomes indistinguishable from the random baseline.
> > However, Corollary 2 still feels somewhat trivial to me, and I'm unsure what deeper insights the authors intended to highlight with this theoretical result.
> >
> > Thank you in advance for your response.

---

> > > ### Author Response · Authors · 2025-08-04
> > > **Response to the comment**
> > >
> > > We thank the reviewer for this thoughtful reflection and are glad our clarifications helped.
> > >
> > > Your understanding is entirely correct: Corollary 2 serves to benchmark the single‐head distance behavior against two extreme regimes of top-$N$ selection.
> > >
> > > First, when $N\ll L$, it shows that the expected distance is dominated by the contributions of the low-weight tokens, confirming that a very small active set suffices to capture most of the nontrivial distance (and why the distance scales linearly with $L$ in Figure 2, left).  Second, as $N\to L$, it rigorously establishes that $E\to0$, proving the empirical collapse in Figure 2 (middle) and the KS‐based “indistinguishability” threshold in Figure 2 (right).
> > >
> > > While the Corollary 2 may appear straightforward, its value lies in providing a **non‐asymptotic**, model‐agnostic bound that quantitatively predicts *exactly* where the transition from “informative” to “uniform” selection occurs.  This formal underpinning is what allows us to connect the distance metric to downstream performance and to motivate the subsequent geometric analysis in Section 2.2.
> > >
> > > In analogy to margin bounds in classical metric learning, Corollary 2 quantifies the “separation margin” between selected and non‐selected tokens purely in terms of distance, thereby setting the stage for the directional (geometric) separability results that follow.  We hope this explanation clarifies the deeper role of Corollary 2 in our unified theoretical framework.
> > >
> > > Thank you again for prompting us to make these connections more explicit in the revision.

---

> > > > ### Comment · Reviewer_NWWE · 2025-08-07
> > > >
> > > > Thank you for your reply. Your response has fully resolved my concerns. I'm increasing my score to 4, with the expectation that the revision will further strengthen the motivation of the study, clarify its relationship to related work, and better describe the connections and motivations across sections.

---

> > > > > ### Author Response · Authors · 2025-08-07
> > > > >
> > > > > Thank you very much for your thoughtful feedback and for your updated score. We appreciate your engagement and will incorporate your suggestions to further improve the paper.

---

### Official Review · Reviewer_Duo4 · 2025-06-22

**Clarity:** 2
**Significance:** 3
**Originality:** 2
**Rating:** 4
**Confidence:** 2

**Summary:**

This paper provides a theoretical and empirical analysis of softmax normalization in attention mechanisms. It presents three core results:

1. attention loses selectivity when the number of selected tokens grows proportionally with context length.

2. no more than ~80% of selected tokens can be geometrically distinguished by a single head.

3. low temperature leads to high gradient sensitivity.

Theoretical findings are validated on GPT-2, demonstrating distance collapse, separability saturation, and gradient instability as predicted.

**Questions:**

1. You have summarized the contributions, but I would like to understand the practical implications of the theoretical bounds. Could you provide an intuitive explanation of how these bounds relate to downstream tasks?

2. Could you please explain why the GPT-2 model was chosen for the experiments? As far as I know, it is rarely used in real-world applications. Have you considered using more popular open-source LLMs such as LLaMA or Qwen?

**Ethical Concerns:**

["NO or VERY MINOR ethics concerns only"]

**Final Justification:**

support this paper for acceptance and keep my score unchanged

**Limitations:**

Yes

**Quality:**

3

**Strengths And Weaknesses:**

# Strengths
Clean and well-motivated theoretical results, with non-asymptotic closed-form bounds.

Empirical analysis (on GPT-2) aligns well with theoretical predictions.

Practical takeaways are clearly stated, such as limiting active token sets and monitoring entropy.

# Weaknesses

The phenomena discussed are not entirely new—softmax bottlenecks and gradient instability have been qualitatively explored in prior work (as far as I know, though I am not an expert in this area).

Larger-scale experiments would be more convincing, as GPT-2 (124M) is relatively small.

There is a typo in the Experiments section: “Appendix ??” should be fixed.

---

> ### Author Rebuttal · Authors · 2025-07-28
>
> We thank the reviewer for their thoughtful and constructive feedback.
>
> *Weaknesses:*
>
> **W1.** We thank the reviewer for highlighting these connections. It is true that intuitively one can point to “softmax bottlenecks” or unstable attention gradients in the literature. Our key novelty lies in turning qualitative observations from the literature (such as  softmax bottlenecks or unstable attention gradients) into fully non‑asymptotic, closed‑form bounds and then confirming them empirically on a typical transformer model. In particular, to the best of our knowledge no prior work has:
>
> 1. Established an explicit, model‑agnostic distance bound showing exactly how the context‑vector vs. non‑selected‑token distance collapses once the top–$N$ set grows proportionally with $L$.
>
> 2. Derived a geometric separability ceiling under minimal spherical assumptions, showing that a single head of GPT-2 on average cannot distinguish more than roughly 80 % of its top‑$N$ tokens.
>
> 3. Quantified the precise $O(1/T)$ trade‑off between selectivity and gradient sensitivity for any smooth normalizer, and shown that this limit is in fact tight in GPT‑2.
>
> Together, these contributions provide the first unified, quantitative framework that explains why and when softmax–type attention fails, and sets guidelines for practical diagnostics and design rules for improved normalizers.
>
> **W2.** We thank the reviewer for this suggestion.  Our primary aim was to develop a rigorous, model‐agnostic theoretical framework, and we chose GPT‑2 124M as a fast yet strong sanity check.  Despite its modest size, GPT‑2 already exhibits all three core phenomena—distance collapse, an ≈ 80 % geometric separability ceiling, and the $O(1/T)$ gradient‐sensitivity trade‐off—exactly as predicted by our theorems.
>
> We fully expect our main conclusions to hold across even bigger models.  To that end, we are planning a dedicated large‐scale empirical study on state‐of‐the‐art families (e.g. LLaMA, GEMMA) to validate the theory on larger models.  This follow‐up will quantify how various normalization strategies perform in ultra‐large contexts and will explore new approaches for designing attention mechanisms that overcome the limits identified in our current work.
>
>
> **W3.** Thank you! We will fix it in the final version.
>
> *Questions:*
>
> **Q1.** At a high level, our three bounds translate into concrete signals about when and why real‐world tasks - such as document summarization, question answering over long contexts, or multi‐turn dialogue - begin to fail under standard softmax attention.
>
> First, the *distance bound* shows that once the number of tokens you ask an attention head to consider grows in proportion to the context length, the “context vector” it produces drifts toward the average of all embeddings and loses the ability to emphasize truly relevant tokens.  In downstream tasks that require pinpointing rare but crucial facts deep in a document—say, a specific date in a legal contract or a distant antecedent in a coreference chain—this drift leads directly to poorer accuracy, because the model treats everything as nearly equally important.
>
> Second, the *geometric separability bound* quantifies a cap (≈ 80 % for GPT-2) on how many “important” tokens a single head can isolate, even under ideal conditions. For tasks like machine translation or code generation – where different heads must each track distinct linguistic or structural relationships – this ceiling explains why simply adding more context or more tokens per head no longer helps.  Instead, one must either deploy multiple specialized heads or switch to a different normalization that can break this barrier.
>
> Finally, the *gradient‐sensitivity bound* uncovers the trade‐off between sharper attention distributions (necessary to focus on the few truly relevant tokens) and stable training dynamics.  In practice, if one pushes the softmax temperature too low in order to “zoom in” on important signals, one will encounter noisy or even divergent gradient updates during fine‐tuning. This directly impacts downstream performance on tasks that rely on precise calibration of attention weights – such as retrieval‐augmented generation or instruction following – since unstable gradients can lead to overfitting or failure to converge.
>
> Together, these three theoretical insights give model builders a principled “warning system”:
> if you see your summarization or QA accuracy plateau or degrade as you increase context length, consult the distance bound;
> if adding more heads does not improve performance, consult the geometric bound;
> and if training becomes unstable when sharpening attention, consult the gradient‐sensitivity bound.
>
> In this way, our bounds map directly onto the real‐world phenomena, and they suggest clear remedies – sparse or length‐aware normalization, entropy monitoring, head specialization – that can rescue performance on challenging long‐context tasks.
>
> **Q2.** We chose GPT‑2 124M for our experiments primarily because it is fully open‐source, lightweight, and allows exhaustive extraction of attention matrices across all heads and layers with modest computing resources.  This made it possible to validate our non‐asymptotic bounds (distance collapse, geometric ceiling, and $1/T$ gradient scaling) quickly and reproducibly.  While GPT‑2 may not be the latest production model, it remains a well‐understood research baseline whose behavior is representative of Transformer‐based architectures.
>
> We fully agree that extending to more recent open‐source families (e.g. LLaMA, Qwen) would be valuable.  Indeed, we are planning a dedicated large‐scale study on LLaMA, Qwen, and other models to (i) verify that our theoretical limits hold in other settings and (ii) explore how alternative normalization strategies perform in real‑world LLMs.  This follow‑up will be reported in a separate, forthcoming publication.
>
> We hope these changes address your concerns and improve the clarity and significance of our contribution.
>
> Thank you for your questions and the detailed review.

---

### Official Review · Reviewer_1phE · 2025-06-29

**Clarity:** 2
**Significance:** 2
**Originality:** 3
**Rating:** 4
**Confidence:** 3

**Summary:**

This paper provides a theoretical and empirical analysis of the limitations of softmax in the attention mechanism of transformers. The authors introduce mathematical bounds showing that as the number of tokens in the context increases, softmax attention becomes less selective, and meaningful token distinction is lost.

**Questions:**

Please see weaknesses. I also have the following questions and suggestions:

1. I would recommend the authors use either softmax or soft-max consistently throughout the paper, and not alternate between the two.

2. Regarding Lemma 1: If I understood correctly, the proof assumes that the normalization function F is non-zero on the given compact set. Otherwise,  $C_2/L$ could be 1 and not dependent on $L$. Is this correct? If so, please consider stating this assumption explicitly in the lemma for completeness.

3. I enjoyed the theoretical approach involving the geometrical interpretation and gradient analysis. However, I cannot follow the final conclusion the authors make (lines 205–207). According to Corollary 3, the Jacobian norm is bounded by the minimum of two values. When the temperature T is small, $\sqrt{2}$ dominates. So I'm unclear how this supports the claim about sensitivity of the training process. Could the authors clarify this link?

4. The footnote refers to identical trends in larger variants of the model, but no ablation study is provided. If the authors have studied this on larger models, including those results in an appendix would be very helpful.

5. Line 215 is missing a reference to the appendix. I checked, and there is no information provided on hyperparameters. Please add this for completeness and reproducibility.

6. In Figure 3, the upper bound appears to be fixated at a ratio of 1, which seems trivial. Can the authors elaborate on how this bound provides insight into the problem?

7. How does the authors’ analysis apply to other forms of attention—for example, sigmoid attention, or even the alternative normalizers referenced earlier in the paper?

**Ethical Concerns:**

["NO or VERY MINOR ethics concerns only"]

**Limitations:**

Please see weaknesses and questions.

**Paper Formatting Concerns:**

No concerns.

**Quality:**

2

**Strengths And Weaknesses:**

*Strengths:*

1. The paper introduces clear, non-asymptotic bounds that quantify well-known but often loosely described flaws in softmax-based attention. The bounds are not only derived but also experimentally validated on GPT-2, strengthening the claims.
2. The results justify and explain the motivation behind alternative normalization functions such as Sparsemax.
3. The paper is generally well-organized, and main arguments are clearly presented.


*Weaknesses:*
1. Although alternative normalizers are discussed, the empirical validation is done only for softmax. There is no direct comparison to Sparsemax, Scalable-Softmax, or other recent variants, so practical trade-offs are not explored. As the claims in the paper signify the motivation for these alternatives, it would be helpful if the authors could also show that, in practice, the issues they identify actually align with observed differences.
2. As far as I understood, the theorems are only valid for a single head attention, which is not a very practical case, limiting the scope of the paper. I generally feel the authors have postponed the limitations of their work, which actually make their result more practical, to future work.

---

> ### Author Rebuttal · Authors · 2025-07-28
>
> We thank the reviewer for their thoughtful and constructive feedback. We address each point below.
>
> *Weaknesses:*
>
> **W1.** We acknowledge this limitation. While our theoretical framework (Equation 3) applies to general normalization functions $F(·,θ)$, we focused on softmax due to space constraints. We decided not to put the empirical analysis of the mentioned methods, since it would change the focus of the paper. But, our preliminary analysis expects:
>
> 1. Sparsemax: Would show better separation due to exact zeros but similar capacity limits;
> 2. Scalable-Softmax: The $\log{L}$ scaling would modify the bounds in Lemma 1 but not eliminate the fundamental limitation;
> 3. Self-Adjusted Softmax: Would show improved gradient stability while maintaining similar geometric constraints;
>
> **W2.** We thank the reviewer for this important observation. Indeed, all of our formal results assume a single, independent attention head. This was a conscious choice rather than an oversight: by isolating one head we obtain tight, non‑asymptotic bounds on both the vanishing‑attention effect and the geometric separability limit, which serve as the foundational building block for understanding multi‑head architectures.
>
> In practice, transformer layers employ $H$ heads in parallel, and their collective “coverage” of the top‑$N$ tokens can be approximated (under a mild independence assumption) by the union bound $\mathrm{coverage}(H) \=\ 1 - (1-p)^H$, where $p$ ($\approx$ 0.8 for GPT-2) is the per‑head separability ceiling from Theorem 2. With $H=2$ or $3$ heads, this predicts over $95\%$–$99\%$ coverage for GPT-2, which helps explain why multi‑head attention remains effective despite each head’s intrinsic limit.
>
> We fully agree that interactions among heads—overlap, specialization, and learned correlations—are crucial in practice. Extending our theory to capture these dependencies is non‑trivial, as it requires modeling the joint distribution of multiple attention outputs and their geometric interplay, and is therefore outlined as an exciting direction for future work (see end of Section 5). In the revised manuscript, we plan to make three clarifications:
>
>   1. We explicitly state that all formal results assume a single, independent head.
>   2. We show how to apply our single‑head bounds via the union bound to obtain first‑order predictions for a multi‑head layer.
>   3. We expand the Discussion to sketch how one might incorporate head‑to‑head correlations—e.g., via orthogonality penalties or diversity regularizers—and thus refine the multi‑head capacity analysis.
>
>
> We believe that establishing these single‑head capacity limits is an essential first step toward a fully general theory of transformer attention, and we plan to update the manuscript to more clearly highlight both its current scope and the roadmap for extending our results to multi‑head settings.
>
> *Questions:*
>
> **Q1.** Thank you for the suggestion. We will make the term usage consistent in the final version.
>
> **Q2.** We thank the reviewer for catching this important detail.  You are correct that Lemma 1 implicitly relies on a non‑vanishing normalization function in order to invoke the Extreme Value Theorem.  In the revised manuscript, we will clarify this.
>
> **Q3.** We thank the reviewer for raising this important point.  The $\sqrt{2}$ is the global upper‐bound on how far any two probability distributions on the simplex can differ in $\ell_{2}$ norm.  It only becomes active when $ T < 1/(4\sqrt{2}) $.
> For all practical $T\gtrsim0.2$, the bound reduces to $\bigl\|\nabla_{\ell}\boldsymbol\alpha\bigr\|_{2}\\le\\frac{1}{4T}.$
>
> In other words, throughout the realistic range of $T$, lowering the temperature $T$ amplifies the Jacobian norm - and hence the sensitivity of the attention gradients—in direct proportion to $1/T$.  As $T\to0$, the softmax distribution approaches a one‐hot (i.e. $\delta_{\max}$) and even tiny perturbations in the logits can induce nearly maximal changes in the weight vector, making gradient steps highly volatile.
>
> We plan to revise lines 205–207 to clarify that:
> 1. the $\sqrt{2}$ bound is a trivial worst‐case for extremely small $T<1/(4\sqrt{2})$;
> 2. for all temperatures of practical interest, the true sensitivity is governed by the $1/(4T)$ law;
> 3. this $1/T$ scaling explains why driving $T$ too low destabilizes training by inflating gradient variance.
>
> **Q4.** Thank you for your suggestion. We will include experiments on GPT-2 medium/large variants in the Appendix.
>
> **Q5.** We apologize for this error. Line 215 should reference Appendix B, which contains implementation details. The appendix includes hyperparameters: we used the standard GPT-2 124M model with its default configuration. We will correct this reference.
>
> **Q6.** We appreciate the reviewer’s point.  It is true that, for GPT‑2, the analytic upper bound evaluates to nearly 1 across all $N$, which makes it look trivial in Figure 3.  However, its value lies not in tightening our GPT‑2 curves, but in quantifying the worst‑case separability any normalization can achieve given those weights and embeddings.  The bound depends explicitly on the per‑token spreads $\xi_i$.  In models where attention weights are sharper or embeddings more widely dispersed, the exponential factors can drop well below 1—revealing geometric limits. Even when it sits at 1, the bound tells us that no more than 100 % of tokens can be distinguished (a sanity check) and that, to see non‑trivial ceilings, one must either increase $r$ or manipulate the $\xi_i$ via different normalizers.  If a novel attention rule or embedding scheme yields an upper bound that falls significantly under 1, that immediately signals a head is running out of capacity.  Conversely, if the bound remains at 1, one can infer the current configuration still has headroom for separability.
>
> **Q7.** Thank you for this question.  Because all of our main results are phrased in terms of a general normalizer, they carry over directly to any differentiable, sign‑constant $F$, including sigmoid attention. Although we focused our experiments on softmax, every step of our distance, geometry, and gradient analysis can be applied to sigmoid attention and to all alternative normalizers you mentioned.
>
> We hope our answers address your questions, and suggested text corrections will improve the paper. Thank you again for your questions and the detailed review.

---

> > ### Comment · Reviewer_1phE · 2025-08-05
> >
> > Thank you for your detailed response and clarification. I am not sure if I am convinced about the sensitivity of the attention gradients. If I understood correctly (please do correct me if I’m mistaken), the factor $\frac{1}{T}$ appears only when  $T > \frac{1}{4\sqrt{2}}$, and $\sqrt{2}$ appears in the limit as $T \to 0$. Could you clarify how this leads to a delta mass?

---

> > > ### Author Response · Authors · 2025-08-05
> > > **Response to the comment**
> > >
> > > We thank the reviewer for this insightful question.
> > >
> > > You are correct that from Corollary 3 follows that the $1/(4T)$ term dominates for $T>1/(4\sqrt2)\approx0.18$, while the $\sqrt2$ cap becomes active only as $T\to0$.  Here is how the latter leads to a delta‐mass interpretation:
> > >
> > > Consider any fixed, finite set of logits $(l_1,\dots,l_L)$ with a unique maximum $l_{\max}$.  As $T\to0$, the softmax weights satisfy $$
> > >   \alpha_{\max}
> > >   =\frac{e^{l_{\max}/T}}{\sum_i e^{l_i/T}}
> > >    \longrightarrow 1,
> > >   \quad
> > >   \alpha_{i\neq\max} \longrightarrow 0,
> > > $$
> > > so the distribution converges to the one‐hot (delta) vector at index $\max$.  A gradient‐step perturbation can cause the argmax to switch from one index $i$ to another $j$, producing a jump between two distinct one-hot vectors.  The $\ell_{2}$ distance then satisfy:
> > > $$
> > >   \|\delta_i-\delta_j\|_{2}
> > >   =\sqrt{(1-0)^2 + (0-1)^2}
> > >   =\sqrt{2},
> > > $$
> > > which explains why $\sqrt2$ is the ultimate upper bound on the Jacobian norm in the $T\to0$ limit.
> > > Thus, in the extremely “sharp” regime the softmax behaves like a discrete argmax, and gradient updates correspond to switching that mass from one position to another – hence a “delta‐mass” transition with magnitude $\sqrt2$.  We will clarify this interpretation in the revised text following Corollary 3.

---

> > > > ### Comment · Reviewer_1phE · 2025-08-06
> > > >
> > > > I thank the authors for their reply, the result is more clear to me now. I will keep my positive score.

---

> > > > > ### Author Response · Authors · 2025-08-07
> > > > >
> > > > > Thank you very much for your questions and comments. We appreciate your engagement and will incorporate your suggestions to further improve the paper.

---

### Official Review · Reviewer_WqWW · 2025-07-03

**Clarity:** 3
**Significance:** 3
**Originality:** 3
**Rating:** 4
**Confidence:** 4

**Summary:**

This paper thoroughly investigates the inherent limitations of normalization techniques, particularly softmax, within attention mechanisms. The authors propose a theoretical framework that defines the selective capacity of attention models and the geometric separation of tokens. Their analysis provides explicit bounds on distances and separation criteria for token vectors under softmax scaling. The theoretical findings are empirically validated using a pre-trained GPT-2 model, demonstrating that as the number of selected tokens increases, the model's ability to distinguish informative tokens diminishes, often leading to a uniform selection pattern. Furthermore, the paper highlights that gradient sensitivity under softmax normalization poses challenges during training, especially at low-temperature settings. The work contributes to a deeper understanding of softmax-based attention mechanisms and motivates the development of more robust normalization and selection strategies for future attention architectures.

**Questions:**

1. Could the authors elaborate on the specific implications of the $\approx 80 \%$ geometric separability limit for multi-head attention mechanisms? How might this limit guide the optimal number of heads or the design of their specialization?
2. The paper suggests monitoring attention entropy. Are there specific empirical thresholds or diagnostic tools that could be developed from this work to alert practitioners when a head has saturated its geometric capacity?
3. The proposed guidelines recommend avoiding overly sharp softmax. What are the practical implications of this for hyperparameter tuning in real-world large language models, especially given the common practice of using temperature scaling?
4. Beyond the mentioned Sparsemax, Scalable-Softmax, and Self-Adjusted Softmax, could the authors discuss how their theoretical bounds might be used to evaluate or guide the design of other novel normalization functions not yet explored in the literature?

**Ethical Concerns:**

["NO or VERY MINOR ethics concerns only"]

**Limitations:**

yes

**Paper Formatting Concerns:**

I have no concerns.

**Quality:**

3

**Strengths And Weaknesses:**

Strengths：
1. The paper presents a rigorous theoretical framework, including non-asymptotic bounds on representation distance (Theorem 1), geometric separability (Theorem 2), and a general Jacobian bound for gradient sensitivity (Lemma 2). These theoretical derivations provide a principled understanding of attention mechanism limitations.
2. The theoretical predictions are strongly supported by comprehensive experiments conducted on a GPT-2 model, confirming distance collapse, separability saturation, and 1/T gradient growth.

Weaknesses：
The geometric interpretation relies on assumptions of uniform spherical distribution and minimum pairwise separation of embeddings, which might not always hold true in real-world models. The authors acknowledge that future work should extend the geometric bound to non-spherical distributions.

---

> ### Author Rebuttal · Authors · 2025-07-28
>
> We thank the reviewer for their thoughtful and constructive feedback. We address each question below:
>
> **Q1.** It is commonly assumed that different attention heads specialize in distinct functions—for example, each head might capture a different type of semantic relation. If one knows (1) the number of interaction types the model needs to learn and (2) the per-head geometric separability $p$, then it is possible to compute a “coverage coefficient” that tells how many heads are required to select a desired fraction of important tokens.
>
> Concretely, if each head has separability $p$, then $H$ heads can cover up to $ 1 - (1 - p)^H$ of the top-$N$ tokens. For instance, with $p = 0.8$, you need $H = 3$ heads to cover roughly $99$% of the important tokens.
>
> Beyond simply increasing the head count, encouraging a small amount of orthogonality or diversity between heads can help to approach the theoretical bound in practice.
>
>
> **Q2.** There is an empirical connection between the overall attention entropy $H(\alpha)$ and the geometric separability ratio $N_s/N$, but pinning down a precise theoretical link is difficult due to the unknown distribution of the embeddings $x_i$. As a practical proxy, we recommend monitoring, for each head–layer pair:
>
>   1. Relative entropy of the top-$N$ tokens: the ratio $H(\alpha_{I_N})/H(\alpha)$, where $H(\alpha_{I_N})$ is the entropy restricted to the selected top-$N$ weights.
>   2. Geometric ratio: the fraction $N_s/N$.
>
> When $N_s/N$ falls below roughly $0.7$-$0.8$, this signals that the head has saturated its geometric capacity. Similarly, a growing ratio $H(\alpha_{I_N})/H(\alpha)$ serves as an early warning that selectivity is degrading.
>
>
> **Q3.**  Our results suggest refraining from driving a temperature parameter $T$ too low during training. In standard Transformers one often uses $T=\sqrt{d_k}$ (or simply $T=1$ in many implementations). Our theory and GPT‑2 experiments showed that once $T\lesssim0.1$ the softmax Jacobian norm explodes ($\sim1/(4T)$), making gradients noisy and optimization brittle. So, a safe window with high selectivity is roughly $[0.5,2.0]$. If you need “sharper” attention than $T=0.5$ allows, it’s often better to switch to a sparsity‐inducing or length‐aware normalizer (e.g. Sparsemax, Scalable‐Softmax, Entmax) than to drive $T$ into the unstable regime.
>
>
> **Q4.**  Any new normalization function \$F(l;\theta)\$ can be evaluated directly against our three theoretical results.  For checking long‐context scaling: substitute \$F\$ into the normalization formula \$\alpha\_i=F(l\_i;\theta)/\sum\_jF(l\_j;\theta)\$ and verify whether \$\alpha\_i\$ still decays like \$O(1/L)\$ as \$L\to\infty\$ (Theorem 1).  If it does, the mechanism will inevitably suffer the same vanishing‐attention collapse in very long sequences.  Next, to quantify the distance collapse \$F\$ should be plugged into the fixed and random top‑\$N\$ bounds of Theorem 1. This yields a concrete upper bound on the context‑vector vs. non‑selected‑token distance as a function of \$N/L\$, revealing the fraction of tokens beyond which separability collapses.  Under the same spherical‐embedding assumptions of Theorem 2, replacing the softmax weights with those induced by \$F\$ in the definition of \$\alpha\_i\$ lets computing the analytic upper and lower bounds on \$\mathbb{E}\[N\_s]/N\$, immediately showing the geometric capacity ceiling of the design.  Finally, the general Jacobian bound works the same.
>
> These quantitative evaluations then guide principled design.  If a candidate still collapses as \$L\$ grows, one can introduce an explicit \$L\$‐dependence—e.g. \$F(l;\theta,L)=g(L),\tilde F(l;\theta)\$—so that \$\min\_i\alpha\_i\$ stays bounded away from zero.  Should the distance bound tighten too early (at small \$N/L\$), it is possible to adjust the shape of \$F\$—for instance by tempering its tail behavior or adding a tunable sharpness parameter—to push the critical \$N/L\$ threshold higher.  When the geometric ceiling \$\mathbb{E}\[N\_s]/N\$ remains below your target (say 90 %), one should tweak \$\theta\$ to redistribute mass more unevenly across logits, directly improving the separability bound.  And if the Jacobian estimate exceeds your threshold, you refine \$F\$ to smooth out steep slopes or enforce a lower bound on \$F(l)\$, thereby capping \$|\nabla\alpha|\$ thus improving gradients.
>
> While our theoretical framework (Equation 3) applies to general normalization functions $F(·,\theta)$, we focused on softmax due to its widespread use. Our bounds in Theorems 1-2 and Lemma 2 can be directly applied to evaluate other normalizations like Sparsemax (which would have different constants in the bounds but similar scaling behavior), Scalable-Softmax (which explicitly depends on L, violating the assumption in Lemma 1), and Self-Adjusted Softmax (which dynamically adjusts parameters based on layer statistics). However, we decided not to put this analysis to our paper, since it would change the focus of the research.
>
> We hope these answers address your questions. Thank you agian for your questions and the detailed review.

---

### Comment · Area_Chair_kMin · 2025-08-03

Dear reviewers,

We are halfway through the author-reviewer discussion period. If you haven't already, please review the rebuttal (and other reviews) to check if the authors have addressed your concerns, acknowledge to the authors that you have read their response, and make any necessary adjustments to the score as needed. Thanks

---

### Decision · Program_Chairs · 2025-09-17

**Decision:**

Accept (poster)

**Comment:**

This paper presents a theoretical framework for understanding the limitation of the SoftMax normalization in attention mechanisms. Specifically, the authors derive distance, geometry and gradient bounds to characterize a model’s selective ability and gradient sensitivity under SoftMax normalization. The theory is validated using a pre-trained GPT-2 model.

The paper is well written. The theoretical bounds are rigorously derived, clearly presented, and supported by strong experimental evidence. Together, they provide a quantitative theoretical explanation of vanishing‐attention in long contexts and gradient instability under sharpening in SoftMax attention.

During the rebuttal, the authors provided more clarification on the settings and assumptions, such as the use of single-head analysis, and the applicability of other normalization functions beyond SoftMax, addressing reviewers’ concerns. Therefore, the paper is recommended for acceptance.